# Epigenetic Mechanisms of LncRNAs Binding to Protein in Carcinogenesis

**DOI:** 10.3390/cancers12102925

**Published:** 2020-10-11

**Authors:** Tae-Jin Shin, Kang-Hoon Lee, Je-Yoel Cho

**Affiliations:** Department of Biochemistry, BK21 Plus and Research Institute for Veterinary Science, School of Veterinary Medicine, Seoul National University, Seoul 08826, Korea; taejin430@snu.ac.kr (T.-J.S.); khlee02@snu.ac.kr (K.-H.L.)

**Keywords:** LncRNA, protein, epigenetic, bind, mechanism, cancer, development, review

## Abstract

**Simple Summary:**

The functional analysis of lncRNA, which has recently been investigated in various fields of biological research, is critical to understanding the delicate control of cells and the occurrence of diseases. The interaction between proteins and lncRNA, which has been found to be a major mechanism, has been reported to play an important role in cancer development and progress. This review thus organized the lncRNAs and related proteins involved in the cancer process, from carcinogenesis to metastasis and resistance to chemotherapy, to better understand cancer and to further develop new treatments for it. This will provide a new perspective on clinical cancer diagnosis, prognosis, and treatment.

**Abstract:**

Epigenetic dysregulation is an important feature for cancer initiation and progression. Long non-coding RNAs (lncRNAs) are transcripts that stably present as RNA forms with no translated protein and have lengths larger than 200 nucleotides. LncRNA can epigenetically regulate either oncogenes or tumor suppressor genes. Nowadays, the combined research of lncRNA plus protein analysis is gaining more attention. LncRNA controls gene expression directly by binding to transcription factors of target genes and indirectly by complexing with other proteins to bind to target proteins and cause protein degradation, reduced protein stability, or interference with the binding of other proteins. Various studies have indicated that lncRNA contributes to cancer development by modulating genes epigenetically and studies have been done to determine which proteins are combined with lncRNA and contribute to cancer development. In this review, we look in depth at the epigenetic regulatory function of lncRNAs that are capable of complexing with other proteins in cancer development.

## 1. Introduction

According to 2018 World Health organization (WHO) statistical data, cancer is the second leading cause of death in the world and accounts for 9.6 million human deaths every year [1]. Cancer has long been studied by many researchers to identify its causes and mechanisms but still, it is unclear how to fully treat cancer and it remains a disease that humanity has not yet conquered [2]. Research for overcoming cancer has been conducted based on the central dogma theory focusing on genes and proteins [3]. Based on the genes and the proteins encoded therein, the interactions between the proteins related to cancer have been discovered and studies have been conducted on the mechanisms of occurrence and metastasis of cancer caused by abnormal mutations of these genes or proteins [4]. Furthermore, beyond only these proteins, over 98% of non-coding transcripts have also begun to attract attention in recent years [5].

Among these non-coding transcripts, long non-coding RNAs (lncRNAs) have a length of 200 bp or more and are epigenetically involved with cancer development in various cells and tissues [6]. LncRNA can play a key role or act as a mediator in epigenetically controlled mechanisms of diverse cancer progression [7]. LncRNAs can regulate oncogenic or tumor suppressive gene expression by directly binding to the target gene or recruiting other transcriptional regulators to induce chromatin modification or DNA methylation in the nucleus [8,9] and by indirectly scaffolding or complexing with miRNA, mRNA, or proteins to control biological functions of the target protein in the cytoplasm [10,11]. Recently, various studies and reviews have focused on the regulatory functions of lncRNA by searching for interactions with miRNAs that regulate target gene expression via binding to its Three prime untranslated region (3′UTR) [12]. However, many lncRNAs can regulate cancer-associated signaling pathways by directly binding to proteins that are related to cancer mechanisms [13]. It has been well-documented that lncRNAs interact with other proteins, including transcription factors (TFs) [14], DNA methyltransferases (DNMTs) [15], polycomb repressive complex 2 (PRC2) [16], RNA binding protein (RBP) [17], and heterogeneous nuclear ribonucleoprotein (hnRNP) [18], to regulate the target’s function in the mechanisms of cancer progression [19]. There have been certain major findings in the mechanisms by which lncRNA cooperates with other proteins: (1) lncRNA interacts with TFs to transcriptionally regulate target gene expression in cancer [14], (2) lncRNA binds to DNMTs such as, DNMT1, DNMT3A, and DNMT3B, causing the alteration of gene methylation [15], (3) lncRNA can remodel chromatin states by forming a complex with PRC2 that is comprised of embryonic ectoderm development (EED), suz12 polycomb repressive complex 2 subunit (Suz12), and methyltransferase enhancer of zeste homolog 2 (EZH2) (Figure 1) [16]. Following these findings, the complexes of various lncRNAs with many different types of proteins have been reported in various types of cancer as well as in different cancer-related processes [20].

In this review, we focused on the lncRNA that functions through direct binding with proteins in the cancer development process, of which the role is further classified as proliferation, migration and invasion, metastasis, apoptosis, microenvironment, and chemoresistance. This will allow us to have a better understanding not only of the association of lncRNA to cancers but also of the molecular mechanisms of target therapeutic proteins. 

## 2. LncRNA–Protein Interaction in Cancer Development

### 2.1. LncRNA in Cancer Proliferation

In the cancer development process, proliferation is a vital start state. The cancer cells are characterized by abnormal proliferation compared to normal cells. Many signal pathways involved in the abnormal cancer cell proliferation have been studied [21]. Here, we summarize that lncRNAs and binding proteins are involved in cell proliferation of several cancers. Breast cancer is the second most common cancer and the fifth most common cause of cancer death in the world [1]. Studies have been reported on the epigenetic regulations of lncRNA in breast cancer. The Nuclear factor kappa-light-chain-enhancer of activated B cells (NF-κB) signaling pathway is known to be activated in the proliferation stages of various cancer cells [22]. LncRNA TROJAN binds to the NF-κB repressing factor (NKRF) protein to interfere with its interaction with the nuclear factor NF-κB p65 subunit (RELA). TROJAN/NKRF increases cyclin-dependent kinase 2 (CDK2) expression to promote proliferation of Estrogen receptor positive (ER+) breast cancer via the G1/S phase transition [23]. Downregulated LINC01585 weakens the binding to the non-POU domain-containing octamer-binding protein (NONO) protein and then NONO interacts with cAMP-response element binding protein (CREB)-regulated transcription captivator (CRTC). Thus, LINC01585 depletion activates cAMP/CREB-related gene expression via recruitment of the NONO/CRTC complex, and promotes the proliferation of breast cancers [24]. In triple-negative breast cancer, ZEB1-AS1/HuR contributes to cancer cell proliferation. LncRNA ZEB1-AS1 binds to ELAVL1 (HuR), a cancer-associated RBP protein, to stabilize Zinc Finger E-Box Binding Homeobox 1(ZEB1) mRNA [25].

Colorectal cancer is the third most common cancer and the second most common cause of cancer death in the world [1]. This cancer also has evidence that is related to lncRNA regulation of its proliferation [26]. Such as LINC01585 binds to HuR in breast cancer, lncRNA OCC-1 also binds to HuR protein and induces ubiquitination through the binding of ubiquitin E3 ligase B-TrCP1 to HuR. OCC-1/HuR complex reduces proliferation of colorectal cancer by repressing cell growth genes, such as hnRNPK and EIF4E, through modulating HuR stability [27]. The hnRNPs, one of the largest families of RBPs, are related to cancer nucleic acid processing by binding to lncRNA. LncRNA Regulator of reprogramming (ROR) increases the half-life of c-Myc mRNA by binding and recruiting with hnRNP1. ROR also binds to AU-rich element RNA-binding protein 1 (AUF1), preventing its interaction with c-Myc mRNA. ROR/hnRNP1/AUF1 proliferates and aggravates colorectal cancer by regulating c-Myc mRNA stability (Figure 2) [28]. Another study suggests that lncRNA SNHG6 binds to EZH2 to methylate the tumor suppressor p21 promoter region and inhibit its expression, which proliferates colorectal cancer [29]. In colorectal cancer, LINC00114 also binds and recruits both EZH2 and DNMT1. It is reported that LINC00114/EZH2/DNMT1 complex upregulates the nucleoporin 214 (NUP214) protein, which is related to mitosis, by decreasing miR-133b expression through hypermethylation of the miR-133 promoter region [30]. The LINC00114 thus eventually help the colorectal cancer cells to proliferate.

Prostate cancer is the fourth most common cancer worldwide [1]. Like breast cancer, prostate cancer is also related in NF-κB signaling and lncRNA. LncRNA PCAT1 interferes with the binding of PHLPP with the FKBPS1/IKKa complex by binding to the complex instead. Thus, PCAT1/FKBPS1/IKKa promotes castration-resistant prostate cancer cell proliferation through enhancing the Protein kinase B (AKT) and NF-κB signaling pathway [31]. Moreover, there are many lncRNAs, which can bind to proteins diversely associated with cell growth, signaling pathways and TFs that can induce proliferative gene expression. LncRNA LOC283070 suppresses the androgen receptor (AR) signaling protein function of prohibitin 2 (PHB2) through direct binding. LOC283070/PHB2 increases cell proliferation through activating AR signaling in prostate cancer [32]. In contrast, lncRNA GAS5 that inhibits prostate cancer directly combines with E2F1 to increase its binding to the P27 promoter. GAS5/E2F1 functions as a tumor suppressor by activating P27 [33]. 

Despite the incidence of gastric cancer being the sixth most common, the mortality rate is the third in the world [1]. Abnormal cell cycle regulation has been well-documented in cancer proliferation [34]. Very recently, several lncRNAs have been revealed to be involved in the cancer cell growth. LncRNA Terminal differentiation-induced non coding RNA (TINCR) transcription is activated by the binding of E2F transcription factor 1 (E2F1) to TINCR’s promoter region. Activated TINCR binds to staufen1 (STAU1) to form a complex and is recruited to the 3′UTR of cyclin-dependent kinase inhibitor 2B (CDKN2B) mRNA, a target of STAU1 and a cell cycle inhibitor, to decay the CDKN2B mRNA. TINCR/STAU1 complex promotes gastric cancer proliferation by increasing the cell cycle as a result of the depletion of CDKN2B [35]. In gastric cancer, FEZF1-AS1, which is accelerated by SP1 transcription factor, recruits the lysine-specific histone demethylase 1A, H3K4me2 demethylase (LSD1), protein to bind at the p21 promoter region and repress its transcription. FEZF1-AS1/LSD1 complex is known to contribute to gastric cancer proliferation via G1/S phase arrest by repressing p21 [36]. Epidermal growth factor receptor (EGFR) is upregulated in numerous cancers [37]. The EGFR-mediated PI3K/AKT signal pathway promotes proliferation of gastric cancer through LINC00152, which directly binds to the EGFR protein [38].

Pancreatic cancer is also reported to be regulated by lncRNAs [39]. LINC01197 and lncRNA SOX2OT regulate proliferation in pancreatic cancer. LncRNA SOX2OT reduces the stability of the FUS RBP protein by binding to it directly. SOX2OT/FUS promotes pancreatic cancer via regulation of cyclin D1 (CCND1) and p27 expression [40]. Aberrant WNT/β-catenin signaling is related to various cancers [41]. LINC01197 prevents β-catenin from binding to TCF4, and directly binds to β-catenin to dysregulate the WNT/β-catenin pathway. Thus, LINC01197/β-catenin complex decreases the proliferation of pancreatic ductal adenocarcinoma by decreasing WNT canonical signaling [42].

Proliferation of other cancers is regulated by lncRNAs. Renal cell carcinoma proliferation depends on the EGFR-mediated pathway which is upregulated by LINC00037/EGFR complex. LINC00037 increases the protein level of EGFR as a result of binding to its protein [43]. Similarly, lncRNA TROJAN, LINC02535, and lncRNA PLAC2 promote cell proliferation via the cell cycle by upregulating CDK2 in renal cancers. To proliferate cervical cancer through CDK2 activation, LINC02535 bound to the poly (rC) binding protein 2 (PCBP2) protein. LINC02535/PCBP2 stabilizes ribonucleotide reductase catalytic subunit M1 (RRM1) mRNA to upregulate CDK2 [44]. Besides, cytoplasmic lncRNA PLAC2 binds to signal transducer and activator of transcription 1 (STAT1) to prevent nuclear transfer. Nuclear PLAC2/STAT1 binds to the ribosomal protein L36 (RPL36) promotor to decrease its expression and inhibit proliferation of glioma via cell cycle arrest by downregulating CDK2 [45]. 

Likes SNHG6 and FEZF1-AS, lncRNA SLNCR commonly target the p21 protein. SLNCR binds to both AR and early growth response 1 (EGR1). SLNCR/AR is recruited to the EGR1-bound promoter of the p21 gene and in turn represses its expression, thereby increasing proliferation of melanoma [46]. Interestingly, along with SNHG6 and LINC00114 targeting p21 via association with EZH2 that was mentioned above, other lncRNA TUG1 can also bind to the EZH2 protein, resulting in the targeting of the CELF1 protein. TUG1 binds to EZH2/EED, which is a subunit of PRC2. Thus, TUG1/EZH2/EED complex inhibits the proliferation of non-small cell lung cancer by binding to the promoter of CUGBP and Elav-like family member 1 (CELF1) and repressing it [47]. In addition, the direct binding of the hepatitis B virus X (HBx) protein to LncRNA DLEU2, resulting in recruiting the PRC2/EZH2 complex to open the chromatin and facilitate an activated state for the target gene, was reported in liver cancer [48]. The above-mentioned lncRNAs that are related to proliferation are summarized in Table 1.

### 2.2. LncRNA in Cancer Migration and Invasion 

Cancer cell migration and invasion are initial stage for metastasis [49]. Many studies support that lncRNA regulates this cancer cellular activity. So, we summarized that lncRNA epigenetic regulation progresses through direct binding to protein in migration and invasion of diverse cancers.

In liver cancer, two lncRNAs GHET1 and CSMD1-1 stabilize activating transcription factor 1 (ATF1) mRNA expression or c-Myc protein respectively, by binding to its protein to regulate cancer cell migration and invasion [50,51]. Involvement of the EZH2 protein in the migration and invasion processes of cancer has also been highlighted with various lncRNAs. Two lncRNA SOX21-AS1 and lncRNA SNHG20 directly interact with EZH2 to recruit and methylate targets promoter region, respectively. SOX21-AS1/EZH2 complex downregulates p21 through directly binding to EZH2 and being recruited to the p21 promoter [52], and SNHG20/EZH2 represses E-cadherin transcription by directly binding to EZH2 [53]. LncRNA SNHG1 is also reported to increase the invasiveness of liver cancer through interacting with DNMT1. SNHG1/DNMT1 represses p53 expression through recruitment to the p53 promoter [54]. 

Also in gastric cancer, downregulated p53 expression contributes to the migration and invasion. LncRNA VCAN-AS1 directly binds to eukaryotic translation initiation factor 4A3 (eIF4A3) to repress p53 [55]. Two lncRNAs interacting with EZH2, lncRNA AFAP1-AS1, and LINC00673 are reported to be involved in the migration of gastric cancers. These two lncRNAs suppress target gene transcription by binding to EZH2. AFAP1-AS1/EZH2 induces epigenetic repression of Krüppel-like factor 2 (KLF2) [56] and LINC00673 can also bind to both EZH2 and lysine demethylase 1A (LSD1), and these complexes decrease large tumor suppressor kinase 2 (LATS2) and KLF2 transcription [57]. Besides, lncRNA SNHG10 recruits DEAD-Box Helicase 54 (DDX54) to maintain PBX homeobox 3 (PBX3) mRNA stability. PBX3 stabilized by SNHG10/DDX54 increases the migration of gastric cancer and activates SNHG10 expression via positive feedback [58].

In glioma, lncRNAs interact with crucial regulatory RBPs like EZH2, DNMT1, and hnRNPL. EZH2 binding with lncRNA NEAT1 that is upregulated via EGFR signaling silences GSK3B, ICAT, and Axin2 at their promoter region. NEAT1/EZH2 complex promotes WNT/β-catenin signaling by increasing the level of nuclear β-catenin during the invasion of glioblastoma [59]. Furthermore, the interaction of DNA methyltransferase (DNMT) proteins with lncRNAs are also frequently involved in the cancer migration and invasion. LINC00467 interact with DNMT1 and stabilize its expression. LINC00467/DNMT1 suppresses p53 expression through recruitment to the p53 promoter, which increases the invasiveness of glioma [60]. Interaction of hnRNPL with lncRNA SChLAP1 prevents the Actinin alpha 4 (ACTN4) protein from proteasomal degradation to stabilize the ACTN4 protein level. SChLAP1/hnRNPL complex enhances NF-κB signaling in invasive glioblastoma through an increase in p65 nuclear localization via ACTN4 [61].

In breast cancer, lncRNA MIAT combines with DNMT1, DNMT3A, and DNMT3B. In migratory, invasive breast cancer, MIAT/DNMT1/DNMT3A/DNMT3B complexes suppress the HIPPO signaling pathway by downregulating discs large MAGUK scaffold protein 3 (DLG3) transcription level via recruitment to the DLG3 promoter region, and it thereby increases the translocation of yes-associated protein 1 (YAP) into the nucleus [62]. Highly expressed lncRNA CCAT1 also binds to Annexin A2 (ANXA2) and glycogen synthase kinase 3 beta (GSK3β), which leads to the nuclear translocation of β-catenin. CCAT1/ANXA2/GSK3β activates the WNT/β-catenin signaling pathway by increasing the amount of nuclear β-catenin, which interacts with transcription factor 4 (TCF4), resulting in the migration and invasion of breast cancer [63].

In other cancers, lncRNAs also have been reported to interact with EZH2, which promote cancer cell migration. First, lncRNA H19, which is upregulated by forkhead box F2 (FOXF2), can recruit the EZH2 protein to bind to the phosphate and tensin homolog deleted on chromosome ten (PTEN) tumor-suppressor gene. H19/EZH2 complex silenced PTEN thus stimulates the migration of non-small cell lung cancer [64]. Second, in migration and invasion of prostate cancer, lncRNA MEG3 suppresses Engrailed-2 (EN2) expression by inducing H3K27me3 on the EN2 promoter, which inhibits the cancer process (Figure 3) [65]. Third, lncRNA DANCR catalyzes H3K27me3 in the Fructose-1, 6- biphosphatase (FBP1) promoter region through binding EZH2. DANCR/EZH2 silences FBP1 and promotes the migration of cholangiocarcinoma [66]. Fourth, binding of lncRNA EBIC to EZH2 enhances cervical cancer invasion through regulating epithelial-to-mesenchymal transition [67]. Fifth, LINC00673 binds to both EZH2 and DNMT1 and recruits them to the p53 promoter. The LINC00673/EZH2/DNMT1 complex increases the invasion of papillary thyroid carcinoma by repressing p53 [68]. 

Additionally, several transcription modulators including TFs, enhancers, and transcription machinery have been found to interact with lncRNAs [14,69]. The binding of lncRNA LEF1-AS1 to hnRNPL elevates the level of lymphoid enhancer binding factor 1 (LEF1) expression. LEF1-AS1/hnRNPL activates migration and invasion in osteosarcoma through stabilization of LEF1 mRNA [70]. In addition, lncRNA HGBC is stabilized by binding with the HuR protein. HGBC/HuR interferes with the binding of SET mRNA with miR502-3p by sponging miR502-3p and thus promotes the migration and invasion of gallbladder cancer though activated AKT signaling [71]. LncRNA SLNCR1, which contains an AR-binding RNA motif, scaffolds AR to upregulate matrix metallopeptidase 9 (MMP9). SLNCR1/AR promotes melanoma invasion by increasing invasive gene activation, such as MMP9 [72]. Binding of lncRNA ELF3-AS1 to Kruppel-like factor 8 (KLF8) increases both ELF3-AS1 expression and KLF8 protein stability. ELF3-AS1/KLF8 stimulates the migration of bladder cancer through increasing MMP9 expression [73]. The above-mentioned lncRNAs that are related to migration and invasion are summarized in Table 2.

### 2.3. LncRNA in Cancer Apoptosis

Apoptosis is literally a cell suicide program that occurs at specific conditions in any living cell. But, there is a significant difference between normal and cancer cell apoptosis. In normal cells, they have a good balance between cell proliferation and apoptosis, but, the problem came from cancer cells. Cancer cells do not want to kill themselves, but they usually set the mechanism to have non-limited cell division and proliferation in any conditions. Many researchers are focused on how cancer suppresses apoptosis and induces proliferation [74]. But, studies on lncRNA roles in apoptosis are still insufficient and need to be further elucidated. So, here, we focused on lncRNAs, and their binding partners play a role in apoptosis.

Interestingly, and similarly to migration and invasion in cancer, many lncRNAs have also been identified that form complexes with EZH2 and DNMT proteins in the apoptosis process of cancer. Song et al. revealed that binding of lncRNA ATB to DNMT1 downregulates p53 gene expression. Thus, ATB/DNMT1 inhibits renal cell carcinoma’s apoptosis by decreasing p53 expression [75]. In bladder cancer, lncRNA DBCCR1-003 also binds to DNMT1, inducing hypermethylation on the promoter region of DBCCR1. DBCCR1-003/DNMT1 complex reduces the expression of DBCCR1 and apoptosis of bladder cancer [76]. Besides, Zhu et al. found lncRNA AWPPH binds to the EZH2 protein to inhibit SMAD4, a tumor suppressor. AWPPH/EZH2 attenuates the apoptosis of bladder cancer by silencing SMAD4 by way of histone methylation on H3K27me3 [77]. 

Like this, many pieces of research have reported that lncRNA binds to EZH2 protein that suppresses apoptosis in osteosarcoma [78], colon cancer [79], pancreatic cancer [80], renal cancer [81], and lung cancer [82]. In hypoxia-induced osteosarcoma, lncRNA FOXD2-AS1, upregulated by HIF1-α, binds to EZH2 and relocates to the promotor region of p21. FOXD2-AS1/EZH2 silences p21 transcription, and thus apoptosis in hypoxia-induced osteosarcoma [78]. In colon cancer, lncRNA ROR1-AS1 can also bind to EZH2 and downregulate the expression of DUSP5. ROR-AS1/EZH2 inhibits the apoptosis of colon cancer by facilitating H3K27me3 on the DUSP5 promoter [79]. In pancreatic cancer, upregulated lncRNA TUG1 binds to and recruits EZH2 to the promoter regions of RND3 and MT2A. Thus, TUG1/EZH2 triggers anti-apoptosis during pancreatic cancer by silencing its tumor-suppressive target genes [80]. In renal cancer, lncRNA HOTTIP recruits both EZH2 and LSD1 to the promoter region of LATS2 and represses LATS2 transcription by increasing H3K27me3. HOTTIP/EZH2/LSD1 downregulates the apoptosis of renal cell carcinoma through silencing LATS2 [81]. In non-small cell lung cancer, similar to lncRNA HOTTIP, lncRNA AGAP2-AS1 reduces apoptosis by interacting with EZH2 and LSD1 to repress LATS2 and KLF2 expression [82]. Collectively, with what was indicated in the migration and invasion in cancer Section 2.2, it can be seen that the dynamic regulation of EZH2 by various lncRNAs might be one of the key regulations in cancer apoptosis and progression. 

Furthermore, several lncRNAs are not related to DNMT or EZH2 proteins but are involved in apoptosis. Thus far, various lncRNAs and protein complexes have been investigated for their anti-apoptotic activity in various cancer types: lncRNA AGAP2-AS1/CBP in breast cancer [83], FMR1-AS1/TLR7 in esophageal squamous cell carcinoma [84], DLEU1/mTOR in endometrial carcinoma [85], FAM83A-AS1/NOP83 in hepatocellular carcinoma [86], APOC1P1-3/tubulin in breast cancer [87], and ZFPM2-AS1/MIF in gastric cancer [88] have been studied by diverse groups. In contrast, downregulated lncRNA P53RRA binds to G3BP1 to displace p53 from it. P53RRA/G3BP1 induces apoptosis in diverse cancers by increasing the nuclear p53 level (Figure 4) [89]. The above-mentioned lncRNAs that are related to apoptosis are summarized in Table 3.

### 2.4. LncRNA in Cancer Metastasis

Metastasis accounts for 90% of cancer deaths and is a process where the specific cells, like cancer stem cells in the primary tumor, travel to nearby or distant organs through lymph nodes or blood vessels, then settle down in a secondary site [90]. Many lncRNAs have been identified that promote metastasis in various cancer types [91]. In this section, we summarize how lncRNA, according to cancer types, binds to its binding partner and is involved in cancer metastasis.

In colorectal cancer, activated NF-κB signaling by binding of lncRNA CYTOR to both NCL and Sam68 promotes metastasis. The CYTOR/NCL/Sam68 trimeric complex enhances Epithelial to mesenchymal transition (EMT) through the NF-κB pathway [92]. Bian et al. identified LncRNA-FEZF1-AS1 and its binding partner. The binding of lncRNA FEZF1-AS1 to PKM2 increases both protein stability and nuclear levels of PKM2, which stimulates STAT3 phosphorylation. FEZF1-AS1/PKM2 promotes the metastasis of colorectal cancer through activation of the STAT3 signaling pathway [93]. Like the above lncRNAs, more lncRNAs were found that bind to proteins in the nucleus. First, the binding of lncRNA RP11 with hnRNPA2B1 downregulates both Siah1 and Fbxo45 expression to stimulate Zeb1 expression. RP11/hnRNPA2B1 complex promotes metastasis in colorectal cancer by increasing the Zeb1 protein level [94]. Second, LINC00858 binds to HNF4a to downregulate WNK2 expression. LINC00858/HNF4a enhances the metastasis of colon cancer by silencing WNK2 [95]. Lastly, lncRNA MALAT1 competitively binds to SFPQ and consequently attenuates its binding to PTBP2. MALAT1/SFPQ activates metastasis in colorectal cancer by releasing PTBP2 [96]. On the contrary, lncRNA GAS5 binds to the YAP protein in cytoplasm and increases its phosphorylation. GAS5/YAP induces the degradation of the YAP protein through ubiquitination and inhibits the transcription of YTHDF3 in normal colorectal cells. But, in colorectal cancer, lower levels of GAS5 triggers the transcription of YTHDF3 by increasing the amount of nuclear YAP proteins. By negative feedback, m^6^A-modification of GAS5 accelerates the metastasis of colorectal cancer through the degradation of GAS5 transcripts that bind to the YTHDF3 protein (Figure 5) [97]. 

In gastric cancer, as in the previous CYTOR, lncRNA KRT19P3 has also been reported to be related to NF-κB signaling. KRT19P3 increases COPS7A stability by directly binding to it. KRT19P3/COPS7A dysregulates the NF-κB signaling pathway through the inhibition of IkBa ubiquitination and thus promotes metastasis in gastric cancer [98]. Besides, lncRNA HOXA11-AS1 binds to both WDR5 and EZH2, to promote β-catenin and repress p21 transcription respectively, and also interacts with STAU1 to degrade KLF2. HOXA11-AS1/WDR5, EZH2, and STAU1 boost metastasis in gastric cancer through regulation of their respective target genes [99]. Conversely, while there are lncRNAs that increase cancer metastasis, there are also lncRNAs that decrease metastasis. LncRNA DRAIC has been known to prevent metastasis. DRAIC induces NFRKB ubiquitination by directly binding to it and thus prevents its binding to UCHL5, which maintains de-ubiquitination of NFRKB. DRAIC/NFRKB reduces gastric cancer metastasis by degrading the NFRKB protein [100].

In Esophageal carcinoma, lncRNA NMR, which is involved in NF-κB, was found by Li and colleagues. LncRNA NMR, which is regulated by NF-κB signaling, binds to BPTF and then activates the ERK1/2 pathway. NMR/BPTF promotes metastasis in esophageal squamous cell carcinoma through the regulation of MMP3 and MMP10 expression via ERK1/2 signaling [101]. In addition, lncCASC9 and its binding partners are reported. Lian et al. found that CASC9 bind to CBP to upregulate LAMC2 expression by complexing at the LAMC2 promoter. CASC9/CBP complex increases LAMC2 expression through histone acetylation (H3K27ac) and promotes pro-metastatic functions in esophageal squamous cell carcinoma [102]. In addition, CACSC9 also binds to EZH2 to enhance metastasis by negatively regulating PDCD4 in esophageal squamous cell carcinoma [103].

LncRNAs are also associated with liver cancer metastasis. LINC00467 binds to the IGF2BP3 protein to stabilize TRAF5 mRNA. LINC00467/IGF2BP3 promotes metastasis hepatocellular carcinoma by retaining TRAF5 mRNA stability [104]. LINC01225 binds to the EGFR protein and elevates its protein level. LINC01225/EGFR complex triggers the EGFR-dependent Ras/Raf-1/MEK/MAPK signaling pathway for metastasis in hepatocellular carcinoma [105].

Chang et al. identified the lncRNA that was involved in STAT3 signaling. LINC00997 binds to the STAT3 protein to relocate to the S100A11 promoter region. LINC00997/STAT3 regulates metastasis molecules, including VIM, MMP2, and MMP7, via S100A11 [106]. Interestingly, Jin et al. reported that the TROJAN from endogenous retrovirus also activates metastasis by suppressing the ZMYND8 protein through inducing degradation by ubiquitination. TROJAN/ZMYND8 activates metastasis-associated genes such as EGFR, VEGFA, and MDM2 in triple-negative breast cancer [107]. LncRNA DANCR binds to the NF90/NF45 complex to stabilize HIF-1α mRNA in metastasis of nasopharyngeal carcinoma [108]. The above-mentioned lncRNAs that are related to metastasis are summarized in Table 4.

### 2.5. LncRNA in Tumor Microenvironment 

Recently, the importance of better understanding the tumor microenvironment has been rapidly increasing [109]. The function of lncRNAs in the tumor microenvironment has been suggested in various processes such as metabolism, angiogenesis, and immune cell infiltration, which are associated with cancer pathology [110,111]. In colorectal cancers, it is reported that lncRNA SNHG6 directly combines with the hnRNPA1 complex, which includes hnRNPA2B1 and PTB, and then increases the proportion of PKM2/PKM1. The SNHG6/hnRNPA1 complex contributes to colorectal cancer metastasis by enhancing aerobic glycolysis [112]. LncRNA SATB2-AS1 binds to WDR5 and GADD45A and increases SATB2 transcription by recruiting them to the SATB2 promoter and mediating H3K4me3. SATB2-AS1/WDR5/GADD45A promotes immune cell infiltration in colorectal cancer via upregulating the SATB2 expression [113]. The binding of lncRNA TNK2-AS1 and STAT3 forms a positive feedback loop that prevents STAT3 proteasomal degradation, and STAT3 in turn activates TNK2-AS1 expression. TNK2-AS1/STAT3 promotes STAT3 signaling, leading to angiogenesis in non-small cell lung cancer via the induction of VEGFA (Figure 6) [114]. 

In breast cancers, lncRNA CamK-A binds to PNCK, which is activated by Ca^2+^ in hypoxia, and IkBa, resulting in its ubiquitination via IkBa phosphorylation. CamK-A/PNCK/IkBa increases the progression of breast cancer microenvironment remodeling through an enhancement in NF-κB signaling [115]. LncRNA HISLA, packaged in extracellular vesicles from tumor-associated macrophages, stabilizes HIF-1α and inhibits degradation by binding to PHD2 in glycolytic tumor cells. HISLA/PHD2 initiates a positive feedback loop and upregulates HISLA in macrophages by releasing lactate [116]. LncRNA NKILA causes activation-induced cell death (ACID) in tumor-specific cytotoxic T lymphocytes (CTLs) and type 1 helper T (T_H_1) cells to promote tumor immune evasion by inhibiting the NF-κB signaling pathway via interaction with p65 and IkBa [117]. The above-mentioned lncRNAs that are related to the tumor microenvironment are summarized in Table 5.

### 2.6. LncRNA in Cancer Chemoresistance

Lastly, several lncRNAs are known to be linked to enhanced drug resistance, and studies on this have recently increased [118]. In breast cancers, it is reported that lncRNA BORG enhances chemo-resistant traits by binding to RPA1, which stimulates the NF-κB signaling pathway in triple-negative breast cancer [119]. Binding of LINC00839 with Lin28B impairs chemotherapeutic outcomes via the PI3K/AKT signaling pathway in breast cancer [120]. LncRNA GBCDRlnc1 binds to PGK1 and prevents its degradation to subsequently activate its downstream targets, such as ATG5-ATG12, that can induce autophagy and chemoresistance in gallbladder cancer [121]. LncRNA LBCS recruits hnRNPK/EZH2 to the SOX2 promoter and represses its expression by inducing H3K27me3. The LBCS/hnRNPK/EZH2 complex promotes chemoresistance in bladder cancer via silencing of SOX2 [122]. 

LncRNA HANR binds to GSKIP to inhibit GSK3β activity, resulting in a reduced effect of doxorubicin on hepatocellular carcinoma [123]. LncRNA PiHL increases drug-resistance by binding to the RPL11/GRWD1 complex and releasing MDM2 from it. PiHL/RPL11/GRWD1 reduces chemosensitivity by allowing for the ubiquitination of p53 by MDM2 in colorectal cancer [124]. LINC01419 combines with DNMT1, DNMT3A, and DNMT3B. The LINC01419/DNMT complex reduces 5-FU sensitivity in esophageal squamous cell carcinoma as a result of this complex binding to the GSTP1 promoter to inhibit its expression (Figure 7) [125]. LncRNA ARHGAP5-AS1 binds to both SQSTM1 and METTL3. ARHGAP5-AS1/SQSTM1 induces autophagic degradation and ARHGAP5-AS1/METTL3 elevates m6A modification of ARHGAP5 mRNA to increase ARHGAP5 transcription. These ARHGAP5-AS1 complexes enhance chemoresistance in gastric cancer [126]. Very recently, the involvement of lncRNA NEAT1 in the drug tolerance of histone deacetylase inhibitors (HDACis) via the suppression of miR-129 has also been documented [127]. The above-mentioned lncRNAs that are related to chemoresistance are summarized in Table 6.

## 3. Conclusions

Although many of these interactions have been reported, only a few studies have presented the structure of lncRNAs and their complexes with proteins [48]. Unlike proteins, the nucleotide sequence of lncRNAs is poorly preserved in evolution. This may indicate that conservation could be accomplished through functional or structural mechanisms rather than through primary sequences. Current research proves that many lncRNAs function together with proteins and other nucleic acids, and that their secondary or tertiary structures are important in mediating these interactions. A variety of trials to predict lncRNA structures have been developed through bioinformatical, biochemical, and three-dimensional (3D) methods, but all have clear limitations in predicting the tertiary structures of lncRNA and its structure when complexed with proteins. The steroid receptor RNA activator (SRA) is the first lncRNA with a characterized secondary structure [128]. Recently, Somarowthu et al. experimentally determined the 3D structure of lncRNA HOTAIR (HOX transcript antisense intergenic RNA) [129]. On the other hand, structural elements such as RNA recognition motif (RRM), ribonucleoprotein particle homology (hnRNPH), zinc-fingers (ZnF), and arginine-rich motifs (ARMs) in the RNA binding proteins have been well studied. Altogether, the mechanisms for lncRNA–protein complexes are still largely unknown, and detailed knowledge of the lncRNA structure and RNA–protein complexes is essential for further implementation.

Through many studies so far, the epigenetic functions of lncRNA and the effects of its interactions with proteins have been studied in diverse disease mechanisms, including cancer development. In this review, we summarized the lncRNAs that are involved in diverse cancer phenotypes via binding to cancer-associated proteins that modulate the subsequent signals (Figure 8). This regulation can occur at the epigenetic level, including remodeling of chromatin states and methylation, as well as the transcriptional level. Of note, the binding of lncRNA to EZH2 protein has been the most frequently demonstrated in various cancer processes. Twenty lncRNAs out of 80 lncRNAs listed in this review interact with EZH2 protein. Six lncRNAs and their functions have been reported with DNMT1 and hnRNPs respectively, which have been well-known as epigenetic modifiers. From another point of view, these various combinations of interactions between lncRNA and proteins regulate cancer phenotypes via modulation of different target proteins. p53 followed by p21 and NF-κB were the target proteins which are the most frequently regulated by various lncRNA complexes. For instance, seven lncRNAs, such as SNHG1, LINC00467, ATB, VCAN-AS1, LINC00673, P53RRA, and ZFPM2-AS1 are involved in cancer process via modulating the p53 signal pathway. In addition, more studies are still required in cancer-associated pathways with other proteins such as HDACS and TLR. Recent studies have provided evidence that several lncRNAs are involved in HDAC and TLR signaling and cancer pathways. Liu et al. identified TLR-related lncRNAs, AP000696.1, LINC00689, LINC00900, and AP000487.1, that are associated with the overall survival of esophageal carcinoma [130], and Chu et al. also reported three lncRNAs (MCHR2, AC011472.4, and AC063944.1) that regulate TLR signaling [131]. Moreover, the involvement of lncRNA H19 [132] and NEAT1 [127] in HDAC activity has been reported in association with cancer. However, the interaction between the lncRNAs and the target proteins have not been well-studied yet. 

Collectively, it is still necessary to discover the functions of new lncRNAs and proteins that have not yet been identified, and to further understand the correlation between these new lncRNAs and proteins in the process of cancer. In addition, research on the structural and physical aspects of the dysregulated interaction of lncRNAs with proteins could be the next step in cancer therapy by way of antibody-mediated cancer treatments or diagnosis kit development utilizing clinical biomarkers. 

## Figures and Tables

**Figure 1 cancers-12-02925-f001:**
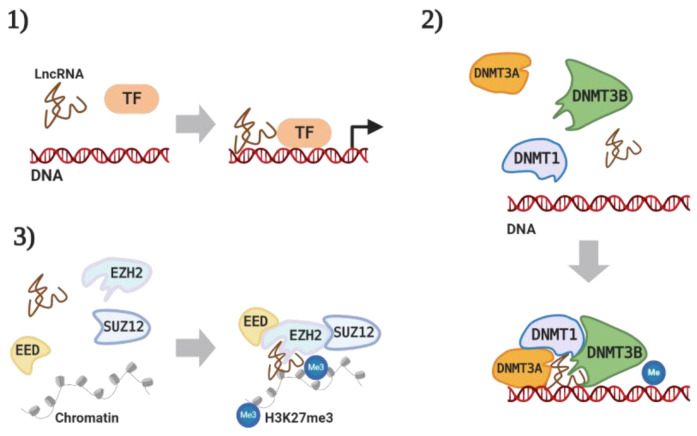
Long non-coding RNA (lncRNA) interaction with RNA binding proteins (RBPs). (**1**) LncRNA regulates gene transcription by binding with transcription factors (TFs), (**2**) lncRNA induces DNA methyltransferases (DNMTs)-mediated methylation, and (**3**) lncRNA complex with polycomb repressive complex 2 (PRC2) to enhance H3K27me3. Figure is created with the program by BioRender.com.

**Figure 2 cancers-12-02925-f002:**
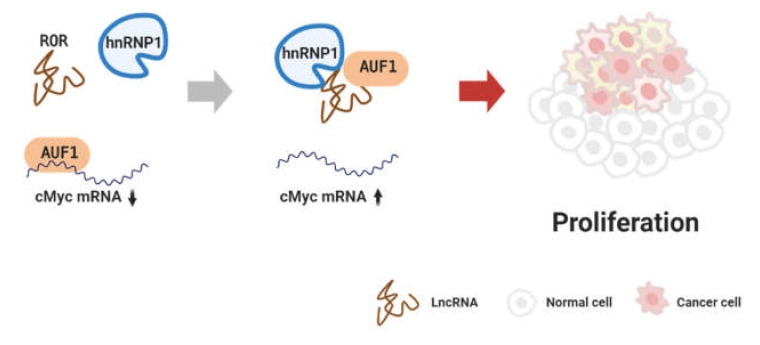
Stabilization of c-Myc mRNA through the lncRNA binding to the AUF1 protein and removing it from the mRNA. The figure was created using BioRender.com.

**Figure 3 cancers-12-02925-f003:**
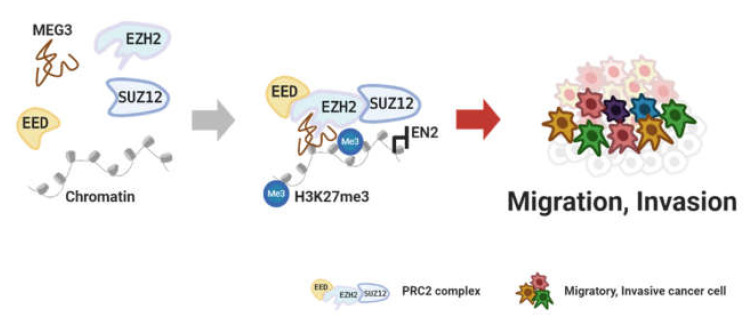
Methylation through lncRNA–protein complex. Figure was created using BioRender.com.

**Figure 4 cancers-12-02925-f004:**
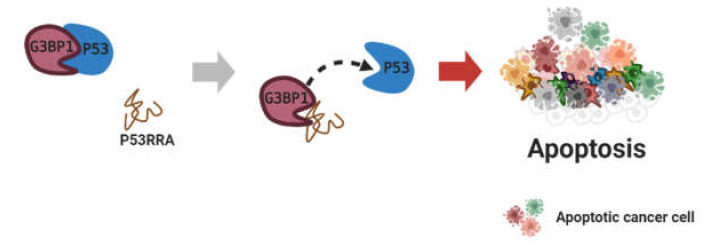
Replacement through lncRNA binding to protein. Figure was created using BioRender.com.

**Figure 5 cancers-12-02925-f005:**
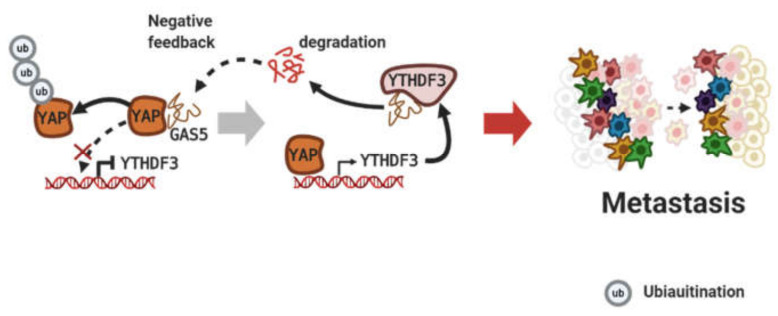
Negative feedback through lncRNA binding to protein. Figure was created using BioRender.com.

**Figure 6 cancers-12-02925-f006:**
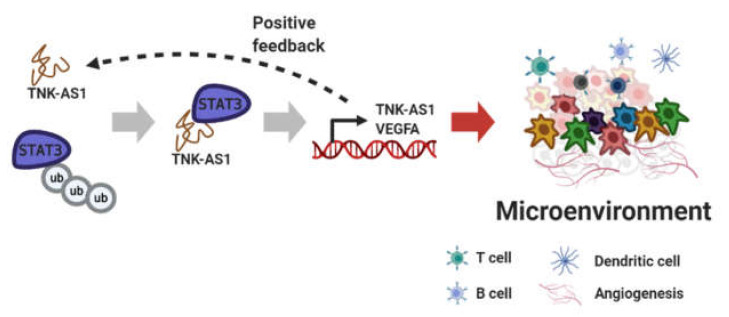
Positive feedback through lncRNA binding to protein. Figure was created using BioRender.com.

**Figure 7 cancers-12-02925-f007:**
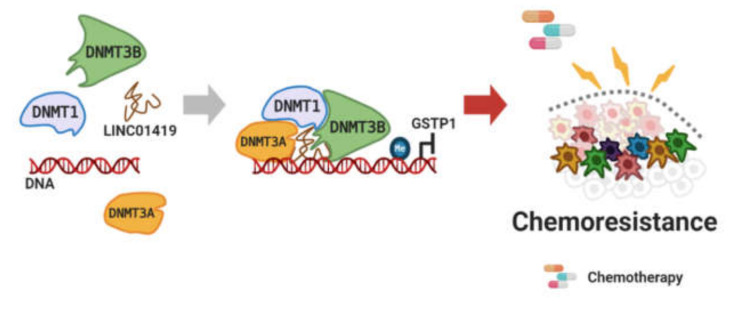
LncRNA binding to DNMTs. Figure was created using BioRender.com.

**Figure 8 cancers-12-02925-f008:**
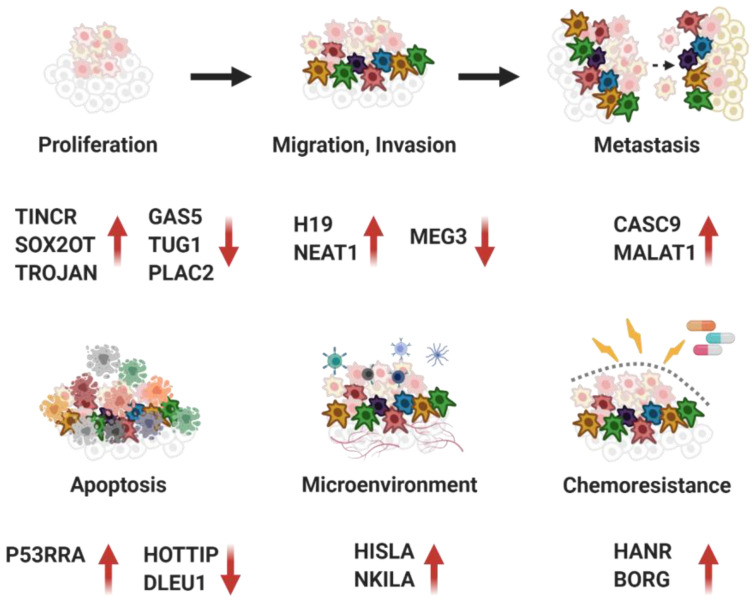
Representative lncRNAs binding to proteins involved in cancer development. Figure is created with the program by BioRender.com.

**Table 1 cancers-12-02925-t001:** Protein-binding lncRNAs involved in cancer cell proliferation.

LncRNA	Binding Protein	Regulatory Protein	Cancer Type	Reference
TROJAN	NKRF	RELA, CDK2	Breast cancer	[23]
LINC01585	NONO	CREB	Breast cancer	[24]
ZEB1-AS1	HuR	ZEB1	Breast cancer	[25]
OCC-1 *	HuR	hnRNPK, EIF4E	Colorectal cancer	[27]
LINC ROR	hnRNP1	c-Myc	Colorectal cancer	[28]
SNHG6	EZH2	p21	Colorectal cancer	[29]
LINC00114	EZH2/DNMT1	NUP214	Colorectal cancer	[30]
PCAT1	FKBP51	NF-κB, AKT	Prostate cancer	[31]
LOC283070	PHB2	AR	Prostate cancer	[32]
GAS5 *	E2F1	P27Kip1	Prostate cancer	[33]
TINCR	STAU1	CDKN2B	Gastric cancer	[35]
FEZF1-AS1	LSD1	p21	Gastric cancer	[36]
LINC00152	EGFR	PI3K/AKT	Gastric cancer	[38]
SOX2OT	FUS	CCND1/p27	Pancreatic cancer	[40]
LINC01197 *	B-catenin	WNT/β-catenin	Pancreatic cancer	[42]
LINC00037	EGFR	EGFR	Renal cancer	[43]
LINC02535	PCBP2	RRM1	Cervical cancer	[44]
PLAC2	STAT1	RPL36, CDK2	Glioma	[45]
SLNCR	AR/EGR1	p21	Melanoma	[46]
TUG1 *	EZH2/EED	CELF1	Lung cancer	[47]
DLEU2	HBx	PRC2	Liver cancer	[48]

* anti-cancer effect.

**Table 2 cancers-12-02925-t002:** Protein-binding LncRNAs involved in cancer cell migration and invasion.

LncRNA	Binding Protein	Regulatory Protein	Cancer Type	Reference
GHET1	ATF1	ATF1	Liver cancer	[50]
CSMD1-1	c-Myc	c-Myc	Liver cancer	[51]
SOX21-AS1	EZH2	P21	Liver cancer	[52]
SNHG20	EZH2	E-cadherin	Liver cancer	[53]
SNHG1	DNMT1	p53	Liver cancer	[54]
VCAN-AS1	eIF4A3	p53	Gastric cancer	[55]
AFAP1-AS1	EZH2	KLF2	Gastric cancer	[56]
LINC00673	EZH2/LSD1	KLF2/LATS2	Gastric cancer	[57]
SNHG10	DDX54	PBX3	Gastric cancer	[58]
NEAT1	EZH2	WNT/β-catenin	Glioma	[59]
LINC00467	DNMT1	p53	Glioma	[60]
SChLAP1	hnRNPL	ACTN4	Glioma	[61]
MIAT	DNMT1/DNMT3A/DNMT3B	DLG3	Breast cancer	[62]
CCAT1	ANXA2	WNT/β-catenin	Breast cancer	[63]
H19	EZH2	PTEN	Lung cancer	[64]
MEG3 *	EZH2	EN2	Prostate cancer	[65]
DANCR	EZH2	FBP1	Cholangiocarcinoma	[66]
EBIC	EZH2	E-Cadherin	Cervical cancer	[67]
LINC00673	EZH2/DNMT1	p53	Thyroid cancer	[68]
LEF1-AS1	hnRNPL	LEF1	Osteosarcoma	[70]
HGBC	HuR	AKT	Gallbladder cancer	[71]
SLNCR1	AR	MMP9	Melanoma	[72]
ELF3-AS1	KLF8	MMP9	Bladder cancer	[73]

* anti-cancer effect.

**Table 3 cancers-12-02925-t003:** Protein-binding lncRNAs involved in cancer cell apoptosis.

LncRNA	Binding Protein	Regulatory Protein	Cancer Type	Reference
ATB	DNMT1	p53	Renal cancer	[75]
DBCCR1-003	DNMT1	DBCCR1	Bladder cancer	[76]
AWPPH	EZH2	SMAD4	Bladder cancer	[77]
FOXD2-AS1	EZH2	p21	Osteosarcoma	[78]
ROR1-AS1	EZH2	DUSP5	Colon cancer	[79]
TUG1	EZH2	RND3, MT2A	Pancreatic cancer	[80]
HOTTIP	EZH2/LSD1	LATS2	Renal cancer	[81]
AGAP2-AS1	EZH2/LSD1	LATS2, KLF2	Lung cancer	[82]
	CBP	MyD88	Breast cancer	[83]
FMR1-AS1	TLR7	NF-κB	Esophageal carcinoma	[84]
DLEU1	mTOR	mTOR	Endometrial carcinoma	[85]
FAM83A-AS1	NOP58	FAM83A	Liver cancer	[86]
APOC1P1-3	tubulin	caspase-3	Breast cancer	[87]
ZFPM2-AS1	MIF	p53	Gastric cancer	[88]
P53RRA *	G3BP1	p53	Lung, Liver, Colon, Nasopharyngeal cancer	[89]

* anti-cancer effect.

**Table 4 cancers-12-02925-t004:** Protein-binding lncRNAs involved in cancer metastasis.

LncRNA	Binding Protein	Regulatory Protein	Cancer Type	Reference
CYTOR	NCL/Sam68	NF-κB	Colorectal cancer	[92]
FEZF1-AS1	PKM2	STAT3	Colorectal cancer	[93]
RP11	hnRNPA2B1	Siah1, Fbxo45	Colorectal cancer	[94]
LINC00858	HNF4α	WNK2	Colon cancer	[95]
MALAT1	SFPQ	PTBP2	Colorectal cancer	[96]
GAS5 *	YAP	YTHDF3	Colorectal cancer	[97]
KRT19P3	COPS7A	NF-κB	Gastric cancer	[98]
HOXA11-AS	WDR5/EZH2/STAU1	KLF2, p21, B-catenin	Gastric cancer	[99]
DRAIC *	NFRKB	UCHL5	Gastric cancer	[100]
NMR	BPTF	MMP3, MMP10	Esophageal carcinoma	[101]
CASC9	CBP	LAMC2	Esophageal carcinoma	[102]
	EZH2	PDCD4	Esophageal carcinoma	[103]
LINC00467	IGF2BP3	TRAF5	Liver cancer	[104]
LINC01225	EGFR	EGFR	Liver cancer	[105]
LINC00997	STAT3	S100A11	Renal cancer	[106]
TROJAN	ZMYND8	EGFR, VEGFA	Breast cancer	[107]
DANCR	NF90/NF45	HIF-1a	Nasopharyngeal carcinoma	[108]

* anti-cancer effect.

**Table 5 cancers-12-02925-t005:** Protein-binding lncRNAs involved in tumor microenvironment.

LncRNA	Binding Protein	Regulatory Protein	Cancer Type	Reference
SNHG6	hnRNPA1	PKM2/PKM1	Colorectal cancer	[112]
SATB2-AS1	WDR5/GADD45A	SATB2	Colorectal cancer	[113]
TNK2-AS1	STAT3	VEGFA	Lung cancer	[114]
CamK-A	PNCK	NF-κB	Breast cancer	[115]
HISLA	PHD2	HIF-1a	Breast cancer	[116]
NKILA	p65/IkBa	NF-κB	Breast cancer	[117]

**Table 6 cancers-12-02925-t006:** Protein-binding lncRNAs involved in cancer chemoresistance.

LncRNA	Binding Protein	Regulatory Protein	Cancer Type	Reference
BORG	RPA1	NF-κB	Breast cancer	[119]
Linc00839	Lin28B	PI3K/AKT	Breast cancer	[120]
GBCDRlnc1	PGK1	ATG5, ATG12	Gallbladder cancer	[121]
LBCS	hnRNPK/EZH2	SOX2	Bladder Cancer	[122]
HANR	GSKIP	GSK3β	Liver cancer	[123]
PiHL	RPL11/GRWD1	p53, MDM2	Colorectal cancer	[124]
LINC01419	DNMT1/DNMT3A/DNMT3B	GSTP1	Esophageal carcinoma	[125]
ARHGAP5-AS1	SQSTM1/METTL3	ARHGAP5	Gastric cancer	[126]

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
