# Peer review of "Epigenetic Mechanisms of LncRNAs Binding to Protein in Carcinogenesis"

_cancers, 2020, doi:10.3390/cancers12102925_

Round 1

Reviewer 1 Report

Comments and suggestions:

The manuscript reviews epigenetic mechanisms of LncRNAs binding to protein in carcinogenesis and their crucial role on proliferation, migration, invasion and metastasis. The review is well written and comprehensive covering the latest advances in the field. However, given the vastly large numbers of published papers and reviews in this field, the authors have discussed only some LncRNAs interaction, which they considered to be the most implicated in carcinogenesis.

They did not clearly separate and distinguish between pro- or anti-cancer effects of LncRNAs interaction in the deceases outcomes. Can some of them be used as biomarkers for the diagnosis and prognosis?

Unfortunately, the authors didn’t discuss the nature and structure of these interactions with the different proteins mentioned. The reader will appreciate to have some information about.

The review is descriptive; we have no detail on the epigenetic mechanisms as stated on the title. Finally, I think that a summary of the clinical and relevant aspects of carcinogenesis therapy that target these interactions will make the review even more interesting.

Reviewer 2 Report

The review article submitted for publication in Cancers and titled "Epigenetic mechanisms of LncRNAs binding to protein in carcinogenesis" describes a recent literature about the topic. The manuscript is well described and emphasizes more aspects related to epigenetic in cancer.

I think that it is suitable for publication in the journal but some modifications should be done before:
-insert more cartoons in the article: they could help the reader to fully understand the topic and create the right conclusions.

-why the role of HDACs and relation with LncRNAs was not treated? although this topic is very new in cancer research, I think that it should be mentioned in the conclusion section to emphasize a perspective role for this review.

-I think that also the role of Toll Like Receptors could be useful for this review. Some evidences involved this family of receptors with LncRNAs.
